# Conic Activation Functions

**Changqing Fu**
CEREMADE
PSL Research University
75016 Paris, France

**Laurent D. Cohen**
CEREMADE
PSL Research University
75016 Paris, France

**Editors:** Marco Fumero, Clementine Domine, Zorah Lähner, Donato Crisostomi, Luca Moschella, Kimberly Stachenfeld

## Abstract

Most activation functions operate component-wise, which restricts the equivariance of neural networks to permutations. We introduce Conic Linear Units (CoLU) and generalize the symmetry of neural networks to continuous orthogonal groups. By interpreting ReLU as a projection onto its invariant set—the positive orthant—we propose a conic activation function that uses a Lorentz cone instead. Its performance can be further improved by considering multi-head structures, soft scaling, and axis sharing. CoLU associated with low-dimensional cones outperforms the component-wise ReLU in a wide range of models—including MLP, ResNet, and UNet, etc., achieving better loss values and faster convergence. It significantly improves diffusion models' training and performance. CoLU originates from a first-principles approach to various forms of neural networks and fundamentally changes their algebraic structure.

## 1 Introduction

Recurrent neural networks (RNNs), convolutional neural networks (CNNs) and Transformers [Vaswani et al., 2017] are examples of a symmetry principle in neural network architectures: they capture local patterns and uniformly apply them across the entire space. These architectures have laid a solid foundation for modern machine learning systems. RNNs repeatedly apply the same weights to the hidden states. This autoregressive form also inspires diffusion models [Sohl-Dickstein et al., 2015]—the patterns are uniform across intermediate states. Convolution layers share the same weights in a small local window to slide across a large domain—the patterns are uniform at arbitrary spatial positions. In Transformers, the self-attention function applies its weights homogeneously to the word or pixel embedding space—the patterns are uniform in arbitrary directions since a per-vector rotation or reflection on both the embedded query and key vectors does not change the attention mask. Different kinds of pattern uniformity are consequences of the associated space homogeneity. These homogeneities (symmetries) have been a principle that continually inspires new designs of model architectures. Recent works continue to push the limit of model performance in vision or language tasks with reduced complexity and different types of symmetry, such as state space models [Gu and Dao, 2023] and more efficient Transformers [Liu et al., 2023].

The convolution and self-attention functions' symmetries are characterized by the equivariance under spatial translation and vector rotation—a function $\lambda$ is equivariant under a group $\mathcal{G}$ if and only if $\forall P \in \mathcal{G}, P\lambda = \lambda P$, where the operation between them is the composition of functions. The same principle applies to a basic multi-layer perceptron (MLP). First, the same activation function is used recurrently in the same space up to a linear embedding layer; second, it applies uniformly to each vector component (neuron). The first property is the foundation of deep models using the same

activation function. The second one results in permutation symmetry: ReLU is equivariant under $\mathcal{G}$ where $\mathcal{G}$ contains compositions of permutations and diagonal matrices with non-negative entries (positive scaling). The symmetry in models is induced by the symmetry of hidden states' space: by substituting the equality $\lambda = P^{-1}\lambda P, \forall P \in \mathcal{G}$ into a two-layer neural network $f(x) = w\lambda(w'x)$, the network stays the same except that the group acts on the weights $(w, w')$ to obtain $(wP^{-1}, Pw')$, which means the order of rows and columns of the weight matrices are exchanged. While permutation symmetry has been a fundamental assumption in neural networks, we take another path to reflect on this axiomatic assumption and raise the question:

*Can forms of equivariance more general than permutation improve neural networks?*

The self-attention function in Transformers positively answers this question. We give a second answer and let activation functions be another solution. To further motivate the activation function, in Appendix G we start from symmetry principles to axiomatically infer the forms of different neural network structures from scratch, where we essentially modify the hypothesis that activation functions are component-wise. We further show in Appendix B that the proposed activation function and the self-attention function share the same type of symmetry, associated with Noether's Theorem. The symmetry group is related to linear mode connectivity explained in Appendix C, meaning that the loss landscape of neural networks is empirically convex modulo the group. Generalizing the group to infinite order fundamentally enlarges the algebraic structure of neural networks.

**Contributions**  We propose Conic Linear Units (CoLU), which introduces orthogonal group symmetry to neural networks. CoLU outperforms state-of-the-art component-wise activation functions such as ReLU in various models including ResNet and UNet for recognition and generation, and keeps the training and inference speed. It achieves remarkable gains in training diffusion models.

## 2  Background

**Component-Wise Activations**  Among the most commonly used activation functions are Rectified Linear Units (ReLU) and its variants, such as Leaky ReLU and Exponential Linear Units (ELU) [Clevert et al., 2015]. There are also bounded ones, such as the sigmoid function or the hyperbolic tangent function used in Hochreiter and Schmidhuber [1997]. In state-of-the-art vision and language models, soft approximations of ReLU are preferred for their better performance, such as Gaussian Error Linear Units (GELU) [Hendrycks and Gimpel, 2016], Sigmoid-Weighted Linear Units (SiLU) [Elfwing et al., 2018], etc. All these functions are component-wise.

**Non-Component-Wise Activations**  Previous works proposing non-component-wise activation functions are essentially different from CoLU, such as using layer normalizations [Ni et al., 2024] or multiplying the input by a radial function [Ganev et al., 2021]. In comparison, CoLU is a generalization of common activations, keeps the favorable conic-projective property unchanged, and improves the performance of neural networks. In the previous version of CoLU [Fu and Cohen, 2024], it had not yet achieved universal improvement on all types of models, since its variants had not been developed.

**Equivariance in Linear Layers**  For symmetries in the *linear* part of the model, ensuring different equivariance improves the performance of recognition [Zhang, 2019] and generation [Karras et al., 2021] models, which repeatedly confirm the potential benefits of the symmetry principle. Group equivariant convolutional neural networks (GCNN) [Cohen and Welling, 2016] put symmetry constraints in the spatial domain so that the model admits spatial group actions such as 2D spatial rotations and reflections. Like in most convolutional neural networks, the channel dimensions of GCNNs are always fully connected. CoLU's symmetry assumption is on the channel axis of the states, which means that CoLU considers the tangent space of GCNN's symmetry space, and equally applies to fully connected layers without convolution structures.

**Spatial versus Channel Correlations**  Invariant scattering convolutional networks [Bruna and Mallat, 2013] use wavelet bases as deterministic spatial correlations and only learn the pixel-wise linear layer or $1 \times 1$ convolution. It indicates that learning channel correlation plays a primary role in representing data patterns compared to spatial connections, and it motivates further investigations

into general symmetries in the channel dimensions—the embedding space. Low-rank adaptation [Hu et al., 2022] and the Query-Key embeddings in the self-attention function are examples of putting low-rank assumptions in the embedding space to represent patterns efficiently. CoLU considers another assumption: it assumes potential subspace orthogonalities.

**Orthogonality in the Embedding Space**  Ensuring orthogonality of the embedding space in the linear layers is twofold. The hard constraint method uses a projection onto the Stiefel manifold during training to ensure the orthogonality of the weights [Jia et al., 2019]. The soft constraint method adds a regularization term to the loss function [Wang et al., 2020] and learns the orthogonality approximately. Orthogonal CNNs outperform conventional CNNs, suggesting that the orthogonality property helps neural networks gain robustness and generalization ability. The self-attention function in Transformers is also orthogonal equivariant. CoLU is compatible with these orthogonal layers to allow layerwise orthogonality in consecutive layers.

**Other Constructions in Nonlinearities**  Weiler and Cesa [2019] conduct a survey on some of the nonlinear functions for equivariant networks, which does not cover the form of CoLU. Liu et al. [2024], Mantri et al. [2024] propose essentially component-wise nonlinearities by leveraging other properties, where the equivariance is still restricted to permutations.

## 3   Conic Activation Functions

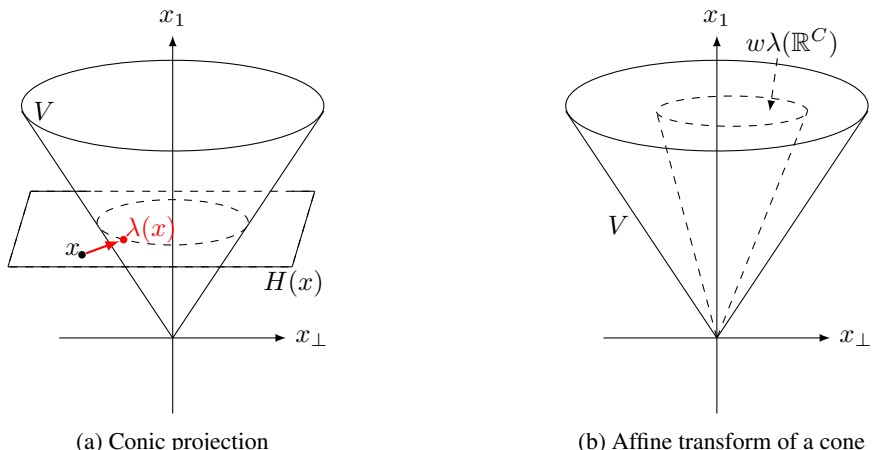

(a) Conic projection                    (b) Affine transform of a cone

Figure 1: Illustration of a CoLU function $\lambda$ and an affine transform $w$ of a cone $V$.

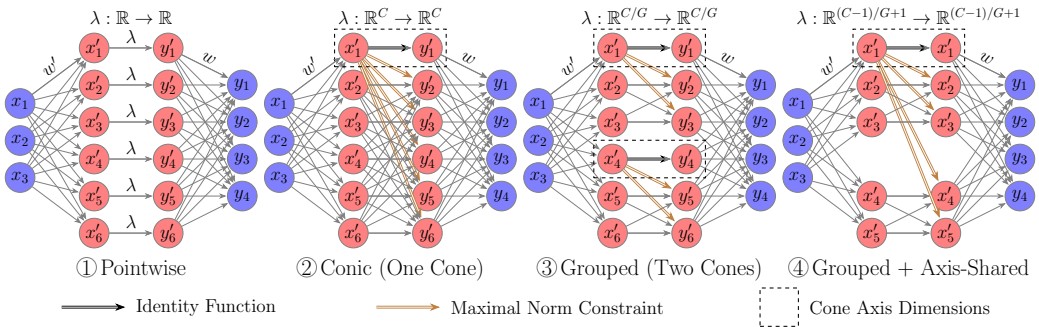

Figure 2: Connections between neurons in a two-layer neural network $y = w\lambda(w'x)$ with component-wise / conic / group-conic / shared-axis group-conic activation functions. In this illustrative example, the network width is $C = 6$ except that in the last shared-axis case $C = 5$. The number of cones is $G = 1$ when there is one cone and $G = 2$ in the grouped case. The yellow arrows denote the maximum norm threshold on the output vector in each group, and the dashed frames denote the cones' axis dimensions.

A basic conic activation function is defined as $\lambda : \mathbb{R}^C \to \mathbb{R}^C$

$$\lambda(x)_i = \begin{cases} x_1, & i = 1 \\ \min\{\max\{x_1/(|x_\perp| + \varepsilon), 0\}, 1\}x_i, & i = 2, \ldots, C \end{cases} \tag{1}$$

where $x = (x_1, x_2, \ldots, x_C)$ is the input vector, $|\cdot|$ is the Euclidean norm, $\varepsilon$ is a small constant taken as $10^{-7}$ for numerical stability, and $x_\perp$ denotes the normal vector $x_\perp = (0, x_2, x_3, \ldots, x_C)$, so that $x = x_1 e_1 + x_\perp$ holds. Here $e_1 = (1, 0, \ldots, 0) \in \mathbb{R}^C$ is a unit vector. Figure 1a visualizes a CoLU function with a red arrow and Figure 1b visualizes a transformed cone with a linear layer after CoLU. Figure 2 visualizes the connections between neurons of the basic CoLU and its variants to be defined in the sequel. The complexity of CoLU is $O(C)$, which is the order of component-wise functions and is negligible compared to matrix multiplications. The design of CoLU is irrelevant to the choice of the first axis or another one since its adjacent linear layers are permutation equivariant.

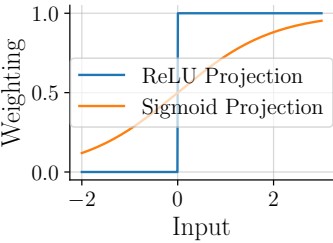

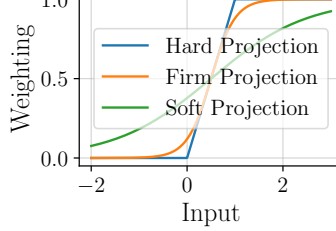

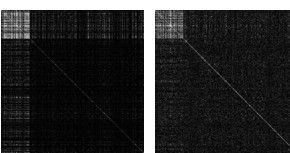

Figure 5: The correlations between weights $\text{cov}(w')$ and between states $\text{cov}(x')$. The bright areas on the top-left corners are the correlated axes.

Figure 3: Weighting of hard-projected ReLU and sigmoid-weighted SiLU.

Figure 4: Weighting of the hard-, firm- and soft-projected conic activation functions.

## 3.1 Soft Scaling

The sigmoid-weighted conic activation function is defined as

$$\lambda(x)_i = \begin{cases} x_1, & i = 1 \\ \text{sigmoid}(x_1/(|x_\perp| + \varepsilon) - 1/2)x_i, & i = 2, \ldots, C \end{cases} \tag{2}$$

where $\text{sigmoid}(x) = 1/(1 + \exp(-x))$. Compared with Equation (1), the weighting function $\min\{\max\{r, 0\}, 1\}$ is replaced by $\text{sigmoid}(r - 1/2)$, where $r = x_1/(|x_\perp| + \varepsilon)$ is the cotangent value of the cone's opening angle $\alpha$, $r \to 1/\tan(\alpha)$ as $\varepsilon \to 0$.

The soft projection is inspired by the better performance of smooth functions such as SiLU $\lambda(x) = \text{sigmoid}(x)x$, compared to the piecewise-linear ReLU $\lambda(x) = \mathbb{1}_{\mathbb{R}_{\geq 0}}(x)x$. Figure 3 compares ReLU weighting with its sigmoid-weighted variant SiLU. Figure 4 compares the hard projection in Equation (1), firm projection weighted by $\text{sigmoid}(4r - 2)$ and sigmoid-weighted soft projection in Equation (2).

## 3.2 Multi-Head Structure

Inspired by group normalization [Wu and He, 2018], group convolution [Krizhevsky et al., 2012], etc., the channel dimension can be divided into $G$ heads of dimension $S = C/G$. The group-conic activation function is defined as a group-wise application of the conic activation function. Suppose $\lambda : \mathbb{R}^S \to \mathbb{R}^S$ is defined in Equation (1) or (2), and $\pi_i^G : \mathbb{R}^C \to \mathbb{R}^S, i = 1, 2, \ldots, G$ are the $G$-partition subspace projections, then $\lambda$ in higher dimension $C$ is uniquely characterized by $\pi_i^G \lambda = \lambda \pi_i^G$, or explicitly,

$$\lambda(x) = (\lambda(\pi_1^G(x)), \lambda(\pi_2^G(x)), \ldots, \lambda(\pi_G^G(x))) \tag{3}$$

In the trivial case $G = 0$, there is no axis to project towards, and we specify that the activation function coincides with the identity function. In the special case $S = 2$ or when the cones are in a 2D space, the 1D cone section degenerates to a line segment with no rotationality, so we specify that the CoLU coincides with the component-wise activation function.

## 3.3 Axis Sharing

The shared-axis group CoLU is also uniquely defined by $\pi_i^G \lambda = \lambda \pi_i^G, i = 1, 2, \ldots, G$ but with the $G$-partition subspace projections defined differently:

$$\pi_i^G = (\pi_1, \pi_{(S-1)(i-1)+2}, \pi_{(S-1)(i-1)+3}, \ldots, \pi_{(S-1)i+1}), \quad i = 1, 2, \ldots, G \tag{4}$$

where $\pi_j, j = 1, 2, \ldots, C$ are projections to each axis. $\pi_i^G$ is a projection onto the first dimension (the cone axis) and $S - 1$ other consecutive dimensions (the cone section). Therefore the relation among the dimension formula among $(C, G, S)$ is $C - 1 = G(S - 1)$ in the shared-axis case.

Figure 5 illustrates the motivation of axis sharing: the colinear effect in the hidden states. In this example, $w'$ is the first linear layer of a VAE's encoder $x \in \mathbb{R}^{784} \mapsto w\lambda(w'x) \in \mathbb{R}^{20}$ pretrained on the MNIST dataset, and $x'$ is the first hidden state $x' = w'x \in \mathbb{R}^{500}$ where the 100 cone axes are permuted together for visualization. Therefore, the hidden dimension is $C = 500$, the number of groups is $G = 100$, the number of test examples is 10000, $w' \in \mathbb{R}^{784 \times 500}$, $x' \in \mathbb{R}^{10000 \times 500}$ and $\mathrm{cov}(w'), \mathrm{cov}(x') \in \mathbb{R}^{500 \times 500}$. The upper-left parts of the matrices are very bright, meaning that the axis dimensions are highly colinear.

## 3.4 Homogeneous Axes

An alternative form of CoLU ensures component homogeneity, by rotating the standard Lorentz Cone towards the all-one vector, and we call it a rotated conic activation function (RCoLU)

$$\lambda(x) = x_e + \max\{\min\{|x_e|/(|x_\perp| + \varepsilon), 0\}, 1\}x_\perp \tag{5}$$

where $x_e = x \cdot e$, $x_\perp = x - x_e$ and $e = (1/\sqrt{S}, \ldots, 1/\sqrt{S})$. The axis-homogeneous cone avoids splitting operations in the calculation. It can also be combined with grouping using Equation (4), and with axis sharing by setting $e = (1/\sqrt{C}, \ldots, 1/\sqrt{C})$ in Equation (5) instead of using Equation (4). RCoLU's performance boost over ReLU is similar to standard CoLU, so we omit it in the experiment section.

# 4 Why Conic Activation Functions

CoLU is motivated by the conic projection, which generalizes the equivariance in a neural network. The proofs are provided in Appendix E.

## 4.1 Conic Projection

To naturally characterize this projection, it is necessary to recall hyperbolic geometry detailed in Appendix A, where we define the Lorentz cone (the future Light Cone) $V = \{x \in \mathbb{R}^C : x_1^2 - x_2^2 - \ldots - x_C^2 \geq 0, x_1 \geq 0\}$ and the hyperplane of simultaneity $H(x) = \{y \in \mathbb{R}^C : y_1 = x_1\}$. We denote $\widetilde{V} = \mathbb{R}_{\leq 0}e_1 \cup V$, where $\mathbb{R}_{\leq 0}e_1 = \{(t, 0, \ldots, 0) \in \mathbb{R}^C : t \leq 0\}$.

**Definition 4.1** (Conic Projection). *The conic projection is defined as $x \in \mathbb{R}^C \mapsto \pi_{\widetilde{V} \cap H(x)}(x)$ where $\pi$ is the nearest point projection, $\pi_A(x) = \mathrm{argmin}_{y \in A} |y - x|$.*

The restriction of the projection on its image $\widetilde{V}$ is the identity function, so it satisfies the idempotent property $\lambda^2 = \lambda$. Constraining the projection in $H(x)$ simplifies the computation while maintaining essential equivariance properties—it guarantees that the projection is always towards the cone axis. Since $V \cap H(x) = \varnothing$ when $x_1 < 0$, the projection is not feasible in the negative half-space, so $V$ is extended to $\widetilde{V}$ for the well-definedness—on the negative half-space, the projection is degenerate, $\pi_{\widetilde{V} \cap H(x)}(x) = (x_1, 0, \ldots, 0)$. In other words, the past Light Cone has zero light speed and thus zero opening angle.

**Lemma 4.2** (CoLU is Conic Projection). *Suppose $\lambda$ is defined in Equation (1), then it coincides with a conic projection.*

$$\lim_{\varepsilon \to 0} \lambda(x) = \pi_{\widetilde{V} \cap H(x)}(x) = \pi_{\max\{x_1, 0\}D + \min\{x_1, 0\}e_1}(x) \tag{6}$$

*where $D = \{x \in \mathbb{R}^C : x_1 = 1, \sum_{i=2}^C x_i \leq 1\}$ is the $(C - 1)$-dimensional disk.*

We note that $V$ is the conic hull of $D$, and $D$ is isometric to a hyperball in dimension $C - 1$, and therefore it has the symmetry group $\mathcal{O}(C - 1)$. In comparison, the invariant set of ReLU is the convex hull of the $C - 1$ simplex $\Delta^{C-1}$, defined as the convex hull of the unit vectors $\{e_i \in \mathbb{R}^C : i = 1, 2, \ldots, C\}$. Next, we discuss the general link between algebraic and geometric symmetry.

## 4.2 Generalized Symmetry Group

Inspired by the Erlangen program [Klein, 1893] bridging algebraic groups with geometric forms, the equivariant group is more intuitively motivated by the symmetry of the projections' invariant sets.

**Definition 4.3** (Invariant Set). *The invariant set of a function $\lambda : \mathbb{R}^C \to \mathbb{R}^C$ is defined as*

$$\mathcal{I}_\lambda = \{x \in \mathbb{R}^C : \lambda(x) = x\}$$

*Moreover, the symmetry group $\mathcal{G}$ and the isometric symmetry group $\mathcal{G}^*$ of a set $A$ is the group of affine and rigid functions that preserves the set:*

$$\mathcal{G}_A = \{P \in \mathrm{GA}(C) : P(A) = A\}, \quad \mathcal{G}_A^* = \mathcal{G}_A \cap \mathcal{E}(n)$$

*where $\mathrm{GA}(C)$ is the general affine group, and $\mathcal{E}(n) = \{P \in \mathrm{Map}(\mathbb{R}^C) : |P(x) - P(y)| = |x - y|, \forall x, y \in \mathbb{R}^C\}$ denotes the Euclidean group.*

**Definition 4.4** (Symmetry Group). *The equivariance group and the isometric equivariance group of a function $\lambda : \mathbb{R}^C \to \mathbb{R}^C$ is defined as*

$$\mathcal{G}_\lambda = \{P \in \mathrm{GA}(C) : P\lambda = \lambda P\}, \quad \mathcal{G}_\lambda^* = \mathcal{G}_\lambda \cap \mathcal{E}(n)$$

**Lemma 4.5** (Projective-Type Operators). *If $\lambda$ is either ReLU or CoLU, then $\mathcal{G}_\lambda = \mathcal{G}_{\mathcal{I}_\lambda}$, and $\mathcal{G}_\lambda^* = \mathcal{G}_{\mathcal{I}_\lambda}^*$.*

This algebra-geometry duality applies to more general neural network architectures, such as the self-attention function. The relation with Noether's theorem is discussed in Appendix B.

**Corollary 4.6** (Permutation Symmetry). *Suppose $\lambda$ is the component-wise ReLU, then $\mathcal{I}_\lambda = \mathbb{R}_+^C$, $\mathcal{G}_\lambda = \mathcal{G}_{\mathcal{I}_\lambda} = \mathcal{S}(C)$ and $\mathcal{G}_\lambda^* = \mathcal{G}_{\mathcal{I}_\lambda}^* = \mathrm{Perm}(C)$, where $\mathbb{R}_+^C = \{x \in \mathbb{R}^C : x_i \geq 0, i = 1, 2, \ldots, C\}$ is the positive orthant, and $\mathcal{S}(C) = \{P\Lambda \in \mathrm{GL}(C) : P \in \mathrm{Perm}(C), \Lambda \in \mathrm{Diag}(C)\}$ is the scaled permutation group in dimension $C$, where $\mathrm{Perm}$ is the permutation group and $\mathrm{Diag}$ is the group of diagonal matrices with non-negative entries.*

**Theorem 4.7** (Conic Symmetry). *The symmetry groups of CoLU defined by Equation (3) or (4) are*

$$\mathcal{G}_\lambda = \mathcal{G}_{\mathcal{I}_\lambda} = \mathcal{S}(G) \times \mathcal{O}^G(S - 1), \quad \mathcal{G}_\lambda^* = \mathcal{G}_{\mathcal{I}_\lambda}^* = \mathrm{Perm}(G) \times \mathcal{O}^G(S - 1) \tag{7}$$

*where $\mathcal{I}_\lambda = \widetilde{V}^G$. In the shared-axis case, $\mathcal{I}_\lambda = \widetilde{V}^G / \sim$ where the relation $\sim$ is defined as $x \sim y$ if and only if $\exists i, j \in \{1, 2, \ldots, G\}$ such that $\pi_i^G(x)_1 = \pi_j^G(y)_1$ and $\forall k \in \{2, 3, \ldots, S\}, \pi_i^G(x)_k = \pi_j^G(y)_k = 0$.*

In Equation (7), $\mathcal{S}(G)$ represents the permutations among different cones and $\mathcal{O}(S - 1)$ represents rotations or reflections within each cone. The motivation is that matrix conjugation modulo permutations reduce to block diagonal form, and we assume there are low-dimensional block sub-spaces that can hold orthogonal equivariance. The symmetry group is continuous and thus of order infinity, unprecedented in component-wise activations. We use the following construction to illustrate that it improves neural networks' generalization ability since component-wise activations fail to hold orthogonal equivariance whereas conic activations do.

**Lemma 4.8** (Layerwise Orthogornal Equivariance). *Assume a two-layer neural network $y = f_\theta(x) = w\lambda(w'x)$ with fixed width $C$ and the training data $\mathrm{D}$ satisfies subspace orthogonal symmetry: $\forall (x, y) \in \mathrm{D}, \forall P \in \mathcal{G}, (Px, Py) \in \mathcal{G}$, where $\mathcal{G} = \{P \in \mathrm{GL}(C) : P[1, 2; 1, 2] \in \mathcal{O}(2), P[3, \ldots, C; 3, \ldots, C] = \mathbf{I}_{C-2}, P[1, 2; 3, \ldots, C] = P[3, \ldots, C; 1, 2]^\top = 0\} \simeq \mathcal{O}(2)$. Then,*

*(1) (ReLU excludes orthogonal equivariance) If $\lambda$ is component-wise activation function, then $\forall \theta \in (\mathbb{R}^{C^2} \backslash \{0\})^2, \exists x \in \mathbb{R}^C$ and $P \in \mathcal{G}$ such that $Pf_\theta(x) \neq f_\theta(Px)$.*

*(2) (CoLU holds orthogonal equivariance) If $\lambda$ is that of Equation (1), then $\exists \theta^\dagger = (w^\dagger, w'^\dagger)$ such that $\forall x \in \mathbb{R}^C, \forall P \in \mathcal{G}, Pf_{\theta^\dagger}(x) = f_{\theta^\dagger}(Px)$.*

As a remark, we explain the sufficiency of rigid alignments with a compact group $\mathcal{G}^*$ without scaling by adding a least-action regularization term, to justify the common practice in the literature, which answers the open issue of permutation-only alignments in Bökman and Kahl [2024].

**Remark 4.9** (Soundness of Isometric Alignment). *Suppose $L$ is the alignment objective defined in the algorithms in Appendix F, then $\exists \eta > 0$ such that the regularized alignment coincides with isometric alignment:* $\operatorname{argmin}_{P \in \mathcal{G}_\lambda}(L(P) - \eta\|P\|) = \operatorname{argmin}_{P \in \mathcal{G}_\lambda^*} L(P)$, *where* $\|\theta\| = -\sum_w (\sum_i w_i^p)^{1/p}$ *is some norm of order $p \geq 1$.*

## 5 Experiments

The experiments are conducted on an 8-core Google v4 TPU. For computational costs, CoLU introduces negligible computational overhead compared to ReLU in all experiments and all variants.

### 5.1 Synthetic Data

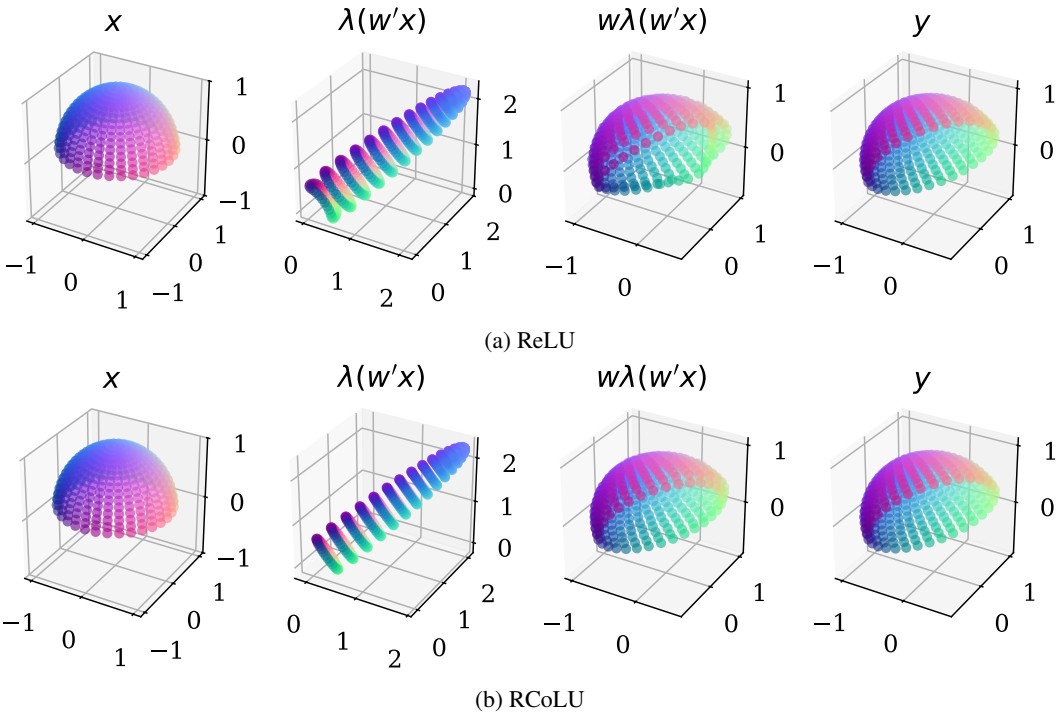

Figure 6: Input, activations, output, and ground truth of a learned hemisphere rotation.

To demonstrate the advantage of the generalized symmetry of CoLU in the embedding space, we use a two-layer MLP to learn the rotation of a 2D hemisphere. The MLP is defined as $x \in \mathbb{R}^3 \mapsto w\lambda(w'x)$, where $w, w' \in \mathbb{R}^{3\times3}$. The dataset D consists of polar grid points and their rotated counterparts $(x, y = Rx)$, where $R$ represents a rotation of $45°$ around each of the three coordinate axes. As shown in Figure 6 in the third column, ReLU does not capture orthogonal equivariance (rotation around the hemisphere axis) near the equator, instead projecting the boundary onto a triangle. In contrast, RCoLU successfully preserves the rotational symmetry at every latitude, including at the boundary. This is due to the geometry of the projection boundary: ReLU cuts the hemisphere with the positive orthant and produces a boundary of the 2-simplex $\Delta^2$, whereas CoLU projects onto a cone that naturally preserves the circular pattern.

## 5.2 Toy VAE

The toy generative model is a VAE with a two-layer encoder and a two-layer decoder, trained on the binarized MNIST dataset. The test loss is compared since CoLU is hypothesized to increase the model's generalization ability.

**Experimental Settings** We use the Adam optimizer with a weight decay of $10^{-2}$ and train 10 epochs for each run. The global batch size is set to 128 and the learning rate is set to $10^{-3}$. Each configuration is trained for 10 times with different random seeds. More detailed settings are provided in Appendix D.

Table 1: Comparisons of CoLU model with soft and hard projections with axis sharing. Unstable means some of the initializations do not converge.

| Width $C$ | Group $G$ | Dim $C$ | Soft? | Train Loss ($\times 10^2$) | Test Loss ($\times 10^2$) |
|---|---|---|---|---|---|
| 2401 | 0 | $\infty$ | Identity | $1.1086 \pm 0.0060$ | $1.1982 \pm 0.0011$ |
|  |  |  | Identity | $1.1072 \pm 0.0031$ | $1.1981 \pm 0.0010$ |
| 2401 | 1 | 2401 | ✓ | $1.0804 \pm 0.0108$ | $1.1740 \pm 0.0009$ |
|  |  |  | ✗ | $1.0835 \pm 0.0048$ | $1.1656 \pm 0.0013$ |
| 2401 | 2 | 1201 | ✓ | $1.0302 \pm 0.0065$ | $1.1216 \pm 0.0016$ |
|  |  |  | ✗ | $1.0226 \pm 0.0057$ | $1.1137 \pm 0.0026$ |
| 2401 | 10 | 241 | ✓ | $0.9181 \pm 0.0060$ | $1.0106 \pm 0.0017$ |
|  |  |  | ✗ | $0.9166 \pm 0.0041$ | $1.0073 \pm 0.0015$ |
| 2401 | 50 | 49 | ✓ | $0.8698 \pm 0.0055$ | $\mathbf{0.9688} \pm 0.0016$ |
|  |  |  | ✗ | $0.8736 \pm 0.0040$ | $\mathbf{0.9742} \pm 0.0024$ |
| 2401 | 200 | 13 | ✓ | $0.8424 \pm 0.0084$ | $\mathbf{0.9643} \pm 0.0015$ |
|  |  |  | ✗ | $0.8430 \pm 0.0052$ | $\mathbf{0.9742} \pm 0.0019$ |
| 2401 | 800 | 4 | ✓ | $0.8388 \pm 0.0268$ | $\mathbf{0.9764} \pm 0.013$ |
|  |  |  | ✗ | Unstable | Unstable |
| 2401 | 1200 | 3 | ✓ | $0.8334 \pm 0.0232$ | $\mathbf{0.9765} \pm 0.0071$ |
|  |  |  | ✗ | Unstable | Unstable |
| 2401 | - | - | SiLU | $0.8429 \pm 0.0034$ | $0.9814 \pm 0.0007$ |
|  |  |  | ReLU | $0.8195 \pm 0.0039$ | $0.9892 \pm 0.0011$ |

Table 2: Comparisons of soft-projected CoLU with or without axis sharing.

| Width $C$ | Group $G$ | Dim $C$ | Share Axis? | Train Loss ($\times 10^2$) | Test Loss ($\times 10^2$) |
|---|---|---|---|---|---|
| 2401 | 0 | $\infty$ | Identity | $1.1086 \pm 0.0060$ | $1.1982 \pm 0.0011$ |
| 2400 |  |  | Identity | $1.1098 \pm 0.0129$ | $1.1985 \pm 0.0015$ |
| 2401 | 1 | 2401 | ✓ | $1.0804 \pm 0.0108$ | $1.1740 \pm 0.0009$ |
| 2401 |  |  | ✗ | $1.0828 \pm 0.0080$ | $1.1733 \pm 0.0008$ |
| 2401 | 2 | 1201 | ✓ | $1.0302 \pm 0.0065$ | $1.1216 \pm 0.0016$ |
| 2402 |  |  | ✗ | $1.0207 \pm 0.0088$ | $1.1179 \pm 0.0029$ |
| 2401 | 10 | 241 | ✓ | $0.9181 \pm 0.0060$ | $1.0106 \pm 0.0017$ |
| 2410 |  |  | ✗ | $0.9111 \pm 0.0041$ | $1.0096 \pm 0.0013$ |
| 2401 | 50 | 49 | ✓ | $0.8698 \pm 0.0055$ | $\mathbf{0.9688} \pm 0.0016$ |
| 2450 |  |  | ✗ | $0.8783 \pm 0.0045$ | $0.9864 \pm 0.0015$ |
| 2401 | 200 | 13 | ✓ | $0.8424 \pm 0.0084$ | $\mathbf{0.9643} \pm 0.0015$ |
| 2600 |  |  | ✗ | $0.8718 \pm 0.0062$ | $0.9833 \pm 0.0021$ |
| 2401 | 800 | 4 | ✓ | $0.8388 \pm 0.0268$ | $\mathbf{0.9764} \pm 0.0139$ |
| 3200 |  |  | ✗ | $0.8801 \pm 0.0073$ | $0.9893 \pm 0.0021$ |
| 2401 | 1200 | 3 | ✓ | $0.8334 \pm 0.0232$ | $\mathbf{0.9765} \pm 0.0071$ |
| 3600 |  |  | ✗ | $0.8808 \pm 0.0099$ | $0.9930 \pm 0.0018$ |
| 2401 | - | - | SiLU | $0.8429 \pm 0.0034$ | $0.9814 \pm 0.0007$ |
| 4800 |  |  | SiLU | $0.8402 \pm 0.0041$ | $0.9856 \pm 0.0008$ |

**Results** Table 1 compares hard-projected or soft-projected CoLU with ReLU or CoLU when the axes are shared. Table 2 compares the improvement from adding axis sharing in the soft projection

case. The test losses at the best early-stopping steps are reported. The highlighted cases correspond to the hyperparameters where CoLU outperforms component-wise activation functions. Furthermore, Appendix D complements the learning curves of these hyperparameters. Combining axis sharing and soft projection effectively stabilizes the training when cone dimensions are low in the VAE experiments.

## 5.3 Toy MLP

According to the hyperparameter search above, we set the cone dimensions to $S = 4$, which complies with the number of chips in hardware platforms. We compare test accuracies in the MNIST recognition tasks to test the hypothesis of CoLU's generalization ability.

**Experimental Settings** We set the global batch size to $1024$ and the learning rate to $10^{-3}$. Each configuration is trained 7 times with different random seeds. More detailed settings are provided in Appendix D.

Table 3: Comparisons between ReLU and CoLU in two-layer MLP.

| Activation | Width $C$ | Dim $S$ | Axis Sharing | Soft Projection | Train Loss | Test Accuracy |
|---|---|---|---|---|---|---|
| ReLU | 512 | - | - | ✗ | 0.0000 ± 0.0000 | 0.9576 ± 0.0017 |
| CoLU | 512 | 4 | ✗ | ✗ | 0.0000 ± 0.0000 | **0.9644** ± 0.0010 |
| CoLU | 511 | 4 | ✓ | ✓ | 0.0000 ± 0.0000 | **0.9652** ± 0.0013 |

**Results** Table 3 compares ReLU with CoLU of low-dimensional orthogonal subspaces and shows the improvement from using axis sharing combined with soft projection.

## 5.4 ResNet

To test the performance of CoLU in deeper models, we scale up the network to ResNet-56 and train them on the CIFAR10 dataset. Axis sharing and soft projection are omitted for clean comparisons with ReLU in the sequel.

**Experimental Settings** The ResNet architecture and the training recipe follow He et al. [2016]. The runs are repeated for 10 times with different random seeds each lasting 180 epochs, and use the Adam optimizer with a batch size of 128, a learning rate of $10^{-3}$, and a weight decay coefficient of $10^{-2}$. Finer training settings will achieve better baselines, and CoLU remains superior to ReLU.

Table 4: Comparisons between ReLU and CoLU in ResNet-56.

| Activation | Cone Dimension $S$ | Train Loss | Test Accuracy |
|---|---|---|---|
| ReLU | - | 0.005132 ± 0.001461 | 0.9065 ± 0.0100 |
| CoLU | 4 | 0.003244 ± 0.000185 | **0.9101** ± 0.0039 |

**Results** Table 4 shows that CoLU outperforms ReLU and the training is stable across different initialization seeds.

## 5.5 Diffusion Models

We compare CoLU and ReLU in unconditional generation with diffusion models [Sohl-Dickstein et al., 2015] trained on the CIFAR10 and Flowers datasets. Then we show the possibility of borrowing a pretrained text-to-image model [Rombach et al., 2022] and fine-tuning it to a CoLU model. Detailed settings are in Appendix D.

**Training Results** Figure 7 shows that CoLU-based UNets converge faster and achieve lower losses than the ReLU-based baselines. On the small dataset CIFAR10, the convergence is observed to be much faster. On the larger Flowers dataset, the loss of the CoLU model is significantly lower

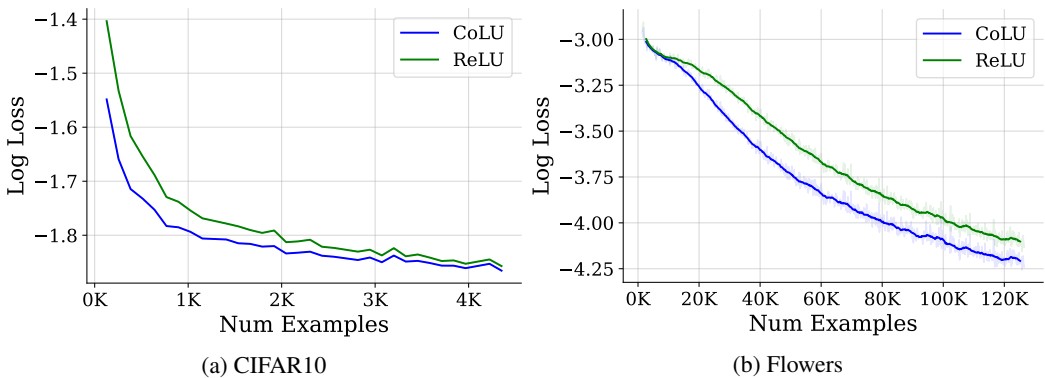

|  (a) CIFAR10  |  (b) Flowers  |

Figure 7: Learning curves of ReLU and CoLU diffusion models.

Table 5: Comparisons between ReLU and CoLU in diffusion UNet.

| Activation | Cone Dimension $S$ | Train Loss (CIFAR10) | Train Loss (Flowers) |
|---|---|---|---|
| ReLU | - | 0.1606 | 0.01653 |
| CoLU | 4 | **0.1593** | **0.01458** |

than the ReLU model throughout the training. Table 5 shows quantitative improvement of CoLU in diffusion UNets. Appendix D shows generated samples on the Flowers dataset.

**Fine-Tuning Results**   We replace all activation functions in the UNet with soft-projected conic activation functions of $G = 32$ without axis sharing. Appendix D shows generated samples from the fine-tuned model and visually compares the original activation and CoLU models.

### 5.6  MLP in GPT2

CoLU is better than ReLU in the MLP part of a Generative Pretrained Transformer (GPT2) trained on Shakespeare's play corpus. Appendix D reports a comparison in the test loss. We also observe that CoLU achieves slower overfitting and lower test loss with the same training loss.

### 5.7  Linear Mode Connectivity

CoLU enlarges the group of neural networks' linear mode connectivity, explained in Appendix C.

**Convolution Filter Symmetry**   Diffusion models with ReLU and CoLU have different symmetry patterns in the convolution filters. We show in Appendix D that between the last layer of two diffusion UNets trained with different initialization on CIFAR10, a ReLU model's convolution filters can be permuted to match each other, whereas a CoLU model cannot since the orthogonal symmetry relaxes to additional color rotations.

**Generative Model Alignment**   For completeness, we show alignment results on the ReLU and CoLU-based models in Appendix D. In the literature on linear mode connectivity, few works study generative models, and we show that the generative VAEs also reveal linear mode connectivity under the equivariance groups of activation functions.

## 6  Conclusion

In this work, we introduced Conic Linear Units (CoLU) to let neural networks hold layerwise orthogonal equivariance. CoLU outperforms common component-wise activation functions and scales to a broad range of large models. The code will be publicly available at `https://github.com/EvergreenTree/di-f-fu-sion`.

## Acknowledgments and Disclosure of Funding

This research is supported with Cloud TPUs from Google's TPU Research Cloud (TRC), and is funded by the PRAIRIE 3IA Institute of the French ANR-19-P3IA-0001 program. We thank anonymous reviewers, Jamal Atif and Antonin Chambolle for their helpful suggestions.

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

# Supplementary Materials

## Contents

**E Proofs**           **21**

**F Algorithms**           **21**

**G Unification of Neural Networks**           **21**

## A Hyperbolic Geometry

**Definition A.1** (Minkowski). *A point (called an event) $x$ is defined in the $C$-dimensional Euclidean space (called space-time). A scalar product on $\mathbb{R}^C$ is defined as*

$$\langle x, y \rangle_M = x_1 y_1 - x_2 y_2 - \ldots - x_C y_C \tag{8}$$

The hyperbolic geometry can be understood by the fact that along a rotation in the space, the quantity $x_1^2 - x_2^2 - \ldots x_C^2$ is unchanged. This scalar product induces a norm $|x|_M = \sqrt{\langle x, x \rangle_M}$, and the Lorentz cone is defined as $V = \{x \in \mathbb{R}^C : |x|_M \geq 0, x_1 \geq 0\}$. It is usually called a *light cone* since if we regard $x_1 := t$ as the time axis where the constant $c$ is the speed of light and $t$ is the time of the event $x$, then the cone is characterized by $\sqrt{|x_\perp|} = ct$, and $c$ is the tangent value of the opening angle of the cone, and we set $c = 1$ without loss of generality. More precisely it is a future light cone since $t \geq 0$, and the past light cone associates to the case when $t \leq 0$. CoLU sets the past light cone with $c = 0$. The *plane of simultaneity* (under the rest frame of reference) is defined as $H(x) = \pi_1^{-1}(x_1 e_1) = \{y \in \mathbb{R}^C : y_1 = x_1\}$. In Figure 1a, CoLU is intuitively understood as the closest point to the input within the light cone and the plane of simultaneity. The meaning of the weight $w$ after the activation function is visualized in Figure 1b, where the previous space-time is tilted by a linear transform (called Lorentz transform). In the grouped CoLU case, gluing the axes together is motivated by equalizing the time axes of each light cone (called an observer).

## B Relation with Noether's Theorem

In this section, we associate the CoLU equivariance with the conserved quantity in the tangent space of the spatial domain and show that CoLU and self-attention have the same type of symmetry.

**Definition B.1** (Lagrangian). *A Lagrangian functional is defined as an integral $\mathcal{L} : TM \longrightarrow \mathbb{R}$ such that*

$$\mathcal{L}(x, \dot{x}, \mathrm{L}) = \int_0^L \mathrm{L}(x, \dot{x}, \ell) \, \mathrm{d}\ell \tag{9}$$

**Theorem B.2** (Noether). *Suppose $\forall s \in \mathbb{R}$ the Lagrangian $\mathcal{L}(x, \dot{x}, L)$ is invariant over a transformation $h^s$ parameterized by $s$, then the following quantity is constant over time.*

$$I = \frac{\mathrm{dL}}{\mathrm{d}\dot{x}} \frac{\mathrm{d}h^s}{\mathrm{d}s} \tag{10}$$

**Corollary B.3** (Translation Momentum). *Assume $\omega \in \Omega = [-1, 1]^2$, $e_1 = (1, 0)$ is a unit vector, and $\mathrm{L}(\omega, \dot{\omega}, t) = \dot{\omega}^2 / 2$. If $h^s(\omega) = \omega + s e_1$ then $I = \dot{\omega}_1$ is conserved.*

The convolution function commutes with $h^s$ and associates with the translation momentum on $\Omega$.

**Corollary B.4** (Angular Momentum). *Assume $x \in \mathbb{R}^\Sigma$ with $\Sigma = \{1, 2, 3\}, |\Sigma| = C = 3$, $e_2, e_3 \in \mathbb{R}^C$ are unit vectors of starting and ending directions of a rotation $R$ around $e_1$. If $h^s : x(\sigma) \in \mathbb{R}^\Sigma \mapsto R^{2s/\pi} x(\sigma)$, then $I = \dot{x} \times e_1$.*

**Proposition B.5** (Attention Invariance). *The self-attention function commutes with $h^s$, so the Lagrangian of attention dynamics admits the orthogonal group. Therefore the attention dynamics in Equation (39) conserves the angular momentum for rotations in $\mathbb{R}^C$.*

**Proposition B.6** (Conic-Activation Invariance). *For the same reason as above, if the activation function is conic, The ResNet dynamics in Equation (40) conserves the angular momentum for rotations around the cone axis.*

## C   Relation with Linear Mode Connectivity

The equivariance of activation functions is linked to the linear mode connectivity phenomenon: two neural networks trained with different initializations and (usually) on the same dataset can be aligned to be very close to each other [Izmailov et al., 2018, Singh and Jaggi, 2020, Entezari et al., 2022, Ainsworth et al., 2023]. This phenomenon implies that neural network optimization is approximately convex modulo a group. The group characterizes the permutation symmetry of component-wise activation functions, and the proposed conic activation functions generalize the type of symmetry. This aligned representation phenomenon across different models at a larger scale is discussed in [Huh et al., 2024]. Note that there are other types of mode connectivity [Garipov et al., 2018], which does not leverage permutation symmetry and requires more complicated paths such as piece-wise linear or Bézier spline, and we do not discuss here.

Given the loss function $L(\theta)$ on two sets of model parameters $\theta_0, \theta_1$, the closeness of the two models is measured by the loss barrier. There are different definitions of loss barriers, and we define it as

$$\sup_{s \in [0,1]} B_{\theta_0, \theta_1}(s) = L((1-s)\theta_0 + s\theta_1)/((1-s)L(\theta_0) + sL(\theta_1)) - 1 \tag{11}$$

The loss barrier signifies the relative loss increase of the linearly interpolated weights. With one model $\theta_0$ fixed, an alignment on the other one $\theta_1$ refers to finding the optimal permutation on each layer by matching either intermediate states or weights [Jordan et al., 2023, Ainsworth et al., 2023]. The proposed activation function generalizes permutations to orthogonal matrices (where permutations are special cases). The orthogonal symmetry is continuous, meaning that there are infinitely many ways of alignment. This results in a loss landscape with infinite local minima forming connected components. The alignment matrices are associated with different manifold constraints.

## D   More Experiments

### D.1   Toy VAE

**Experimental Settings**   The VAE's encoder and generator's parameters are $\theta_E = (w_E, w'_E)$ and $\theta_G = (w_G, w'_G)$. The inputs, latents and outputs are $x, z, \widehat{x}$, where $z = w_E \lambda(w'_E x)$ and $\widehat{x} = w_G \lambda(w'_G z)$. The dimension of input and output is $28 \times 28 = 784$ and the dimension of the hidden state $z$ is fixed to $d = 20$. The loss function is defined as

$$L(\theta) = H(x, \widehat{x}) + \alpha \mathcal{D}_{KL}(p_z | p_0) \tag{12}$$

where $H(x, \widehat{x}) = -\sum_n x_n(\log(\widehat{x_n}) + (1 - x_n)\log(1 - \widehat{x_n}))$ is the binary cross-entropy, and $\mathcal{D}_{KL}(p_z | p_0)$ is the Kullback-Leibler Divergence from a standard Gaussian distribution $p_0 \sim \mathcal{N}(0, 1)$ to the latent distribution $p_z \sim \mathcal{N}(\mu_z, \sigma_z)$

$$\mathcal{D}_{KL}(p_z | p_0) = -\int_x p_0(x) \log(p_z(x)/p_0(x)) \, \mathrm{d}x = \frac{1}{2} \sum_{j=1}^{d} \left(1 + \log(\sigma_{zj}^2) - \mu_{zj}^2 - \sigma_{zj}^2\right) \tag{13}$$

The last equality is obtained by setting $\mu_z = (\sum_{n=1}^{N} z_n)/N$ and $\sigma_z = ((\sum_{n=1}^{N}(z_n - \mu_z)^2)/(N - 1))^{1/2}$ with sample size $N$. The hyperparameter $\alpha$ is set to 1 so that the impact of the KL term is relatively small, given that the magnitude of the cross-entropy term is around 100 times larger.

**More Results**   Figure 8 visualizes the test loss curves when the granularity of grouping varies. In summary, as the cone dimension $S$ reduces, the performance of grouped conic activation functions improves until it outperforms component-wise activation functions. In the figures, only one case for high and low dimensional cones is shown for clarity. The cone dimension is among $S \in \{\infty, 2400, 600, 4, 2\}$, or equivalently, the number of groups is among $G \in \{0, 1, 4, 800, 2400\}$. In the shared-axis case, the network width is fixed to $C = 2401$ so that the number of parameters is the same. In the no-sharing case, the network width is $C = 2400 + G$. Specifically, $G = 0$ reduces to the identity function, while $S = 2$ (or $G = 2400$) is specified as the component-wise activation of ReLU for hard projection and SiLU for soft projection since there is no orthogonality. The dashed line in the hard-projected shared-axis case means that the training is unstable over different random seeds: about $80\%$ of the initializations do not converge, so only one converged training instance is

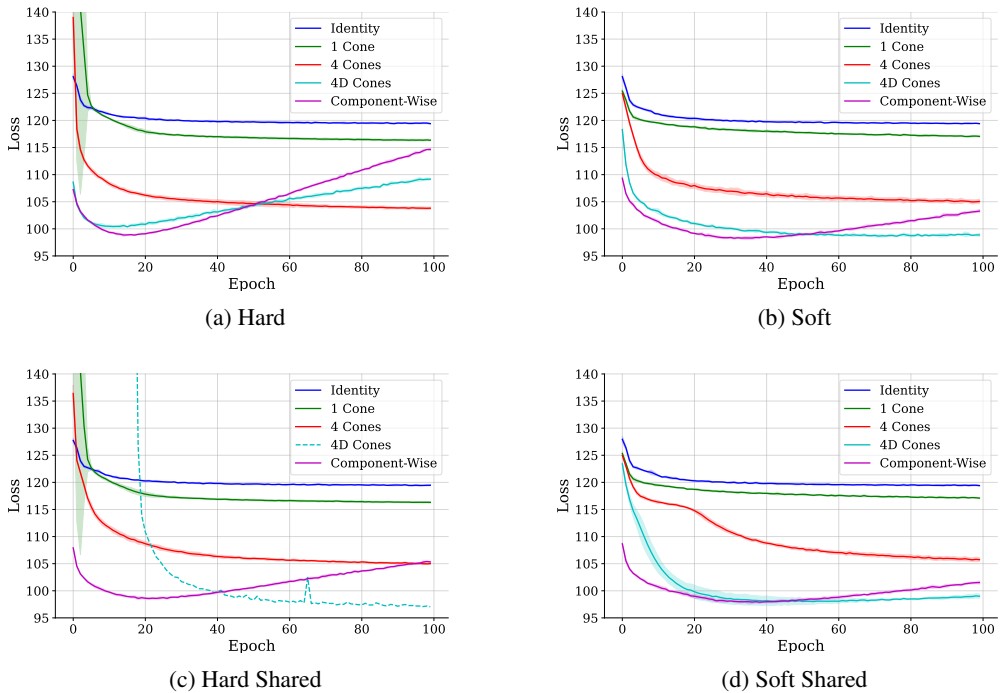

(a) Hard

(b) Soft

(c) Hard Shared

(d) Soft Shared

Figure 8: Test loss curves of a VAE with two-layer encoder and decoder with standard deviation regions. The left and right figures correspond to hard and soft projections, and the top and bottom correspond to hard projection and soft projection.

visualized. Sharing the axes increases the performance of the activation function on toy examples when the projection is soft and the cones are low dimensional ($S$ being small). In practice, this combination is the most meaningful one since it has the best performance and saves the most number of parameters. Intuitively, soft projection effectively stabilizes the training of CoLU models, which is the most obvious in the early training stage of the highest-dimensional conic functions (the single cone case). Especially, it makes the VAE with shared-axis activation functions easier to train.

## D.2 Toy MLP

**Experimental Settings** The model is parameterized by $\theta = (w, w')$ and defined as $x \in \mathbb{R}^{28 \times 28} \mapsto \widehat{y} = \mathrm{softmax}(w\lambda(w'x)) \in \Delta^9$, which is a two-layer MLP whose output is mapped to the probability simplex by a softmax function. The MNIST dataset is denoted as a collection of data pairs $(x, y)$, where $x$ is flattened as vectors and $y$ is a unit vector among 10 classes. The network width is fixed to $C = 512$. The loss function is the cross entropy of the predicted probability relative to the label

$$H(\widehat{y}, y) = \sum_i y_i \log \widehat{y}_i \tag{14}$$

## D.3 Diffusion Models

**Training Experimental Settings** The UNet structure follows the Stable Diffusion model (LDM) [Rombach et al., 2022] without the VAE part. The network block widths are set to $(128, 256, 256, 256)$ and the numbers of ResNet blocks are set to 1 for CIFAR10 (2 for Flowers). For unconditional generation, the cross-attention function is replaced with the self-attention function. All runs last 100K steps and use the Adam optimizer with a batch size of 128 for CIFAR10 (16 for Flowers), a learning rate of $10^{-4}$, and a weight decay coefficient of $10^{-2}$. Figure 9 shows comparisons on the Flowers dataset.

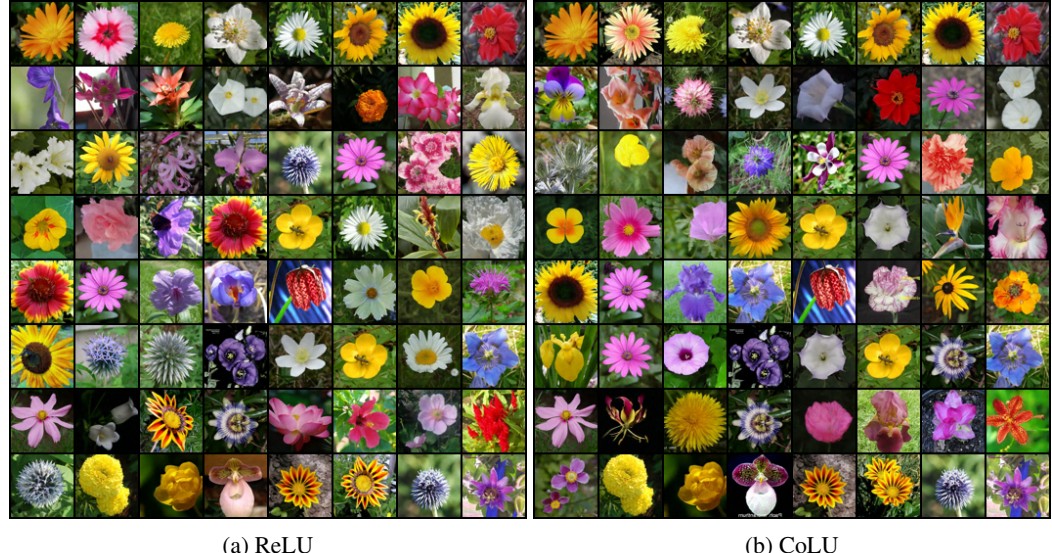

(a) ReLU                                                    (b) CoLU

Figure 9: Samples of diffusion models trained on the Flower Dataset.

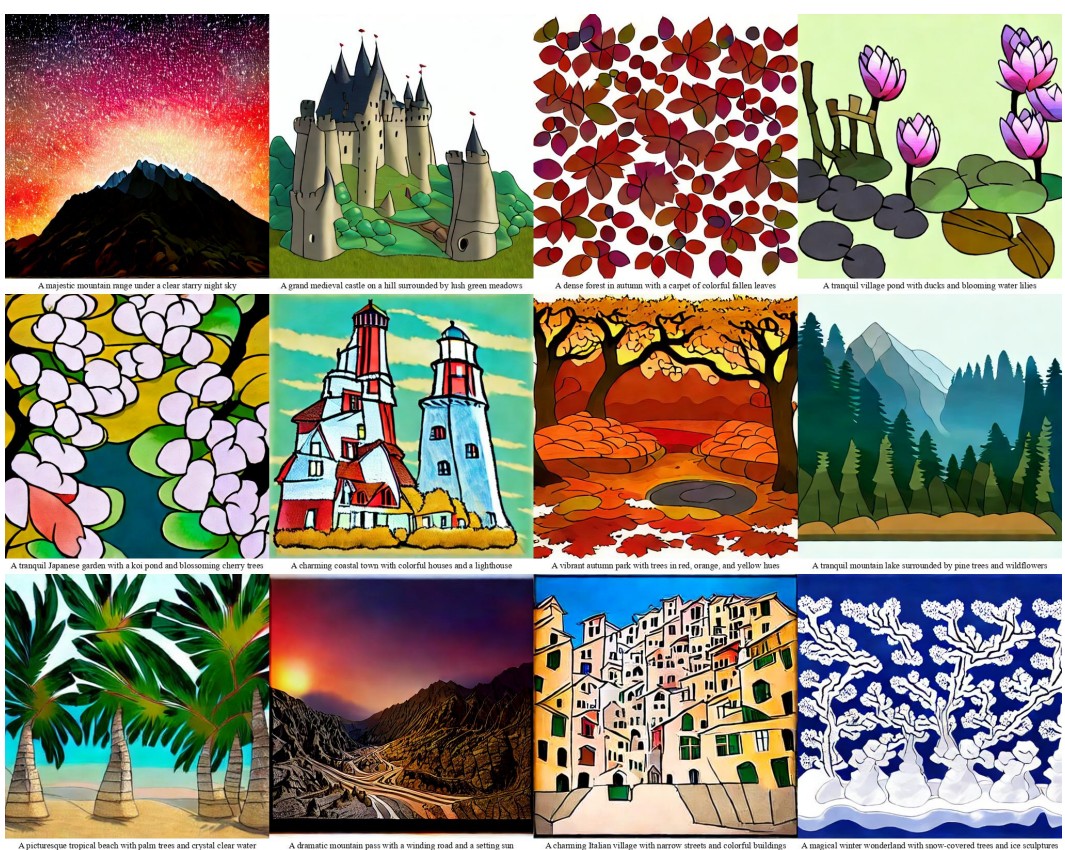

Figure 10: More CoLU text-to-image samples.

**Fine-Tuning Experimental Settings** The pretrained model has 835 million parameters and is trained on the LAION dataset. The architecture is identical to the Stable Diffusion model with block width $(320, 640, 1280, 1280)$. The training details are the same as above. The pre-trained SiLU model and the text-to-image Pokémon dataset are from the diffusers library [von Platen et al.]. Figure 11 visualizes the comparisons between a fine-tuned SiLU model and a fine-tuned soft CoLU

model with the same text prompt and initial noise in the diffusion model. Figure 10 shows more samples of the fine-tuned model with text prompts generated by a large language model.

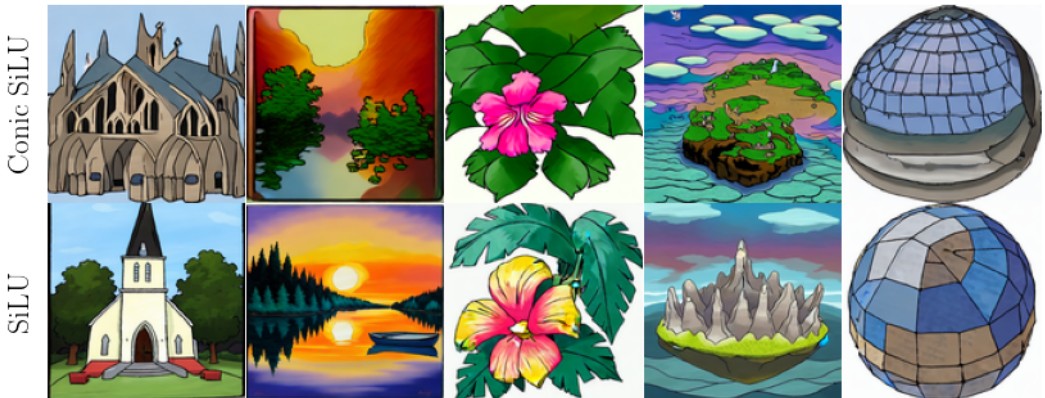

Figure 11: LDM samples of a fine-tuned Soft CoLU model and a fine-tuned SiLU model.

## D.4  MLP in GPT2

**Experimental Settings**  The Transformer follows Vaswani et al. [2017] with the block size of $64$, an embedding size of $256$, a number of heads $8$, head size $32$ and number of layers $6$. Each run lasts 20K steps and uses the Adam optimizer with a batch size of $512$, a learning rate of $10^{-4}$, and a weight decay coefficient of $10^{-2}$.

Table 6: Comparisons between ReLU and CoLU on GPT2's MLP.

| Activation | Cone Dimension $S$ | Train Loss | Eval Loss |
|---|---|---|---|
| ReLU | - | 1.256 | 1.482 |
| CoLU | 4 | 1.263 | **1.481** |

**Results**  Table 6 shows that CoLU is on par with ReLU in GPT2's MLP. We also observe a faster drop in the test loss and slower overfitting.

## D.5  Linear Mode Connectivity

The latent state's permutation symmetry is studied qualitatively on diffusion UNets and quantitatively on toy models.

**Convolution Filter Symmetry**  We train individual diffusion UNets on the CIFAR10 dataset with different random seeds and qualitatively show that the palette filters (the last convolution layer of the

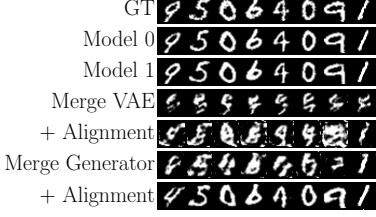

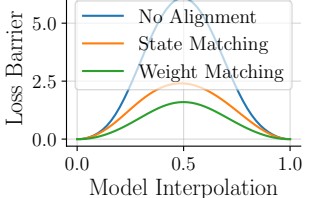

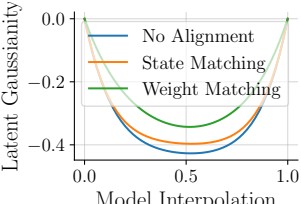

Figure 12: Random samples of ground truth, VAE outputs, merged VAE, and merged generator with fixed latents.

Figure 13: Loss barriers between the aligned models by state or weight matching.

Figure 14: KL Divergence between $\mathcal{N}(\mu_z, \sigma_z)$ and $\mathcal{N}(0, 1)$ on the interpolation paths.

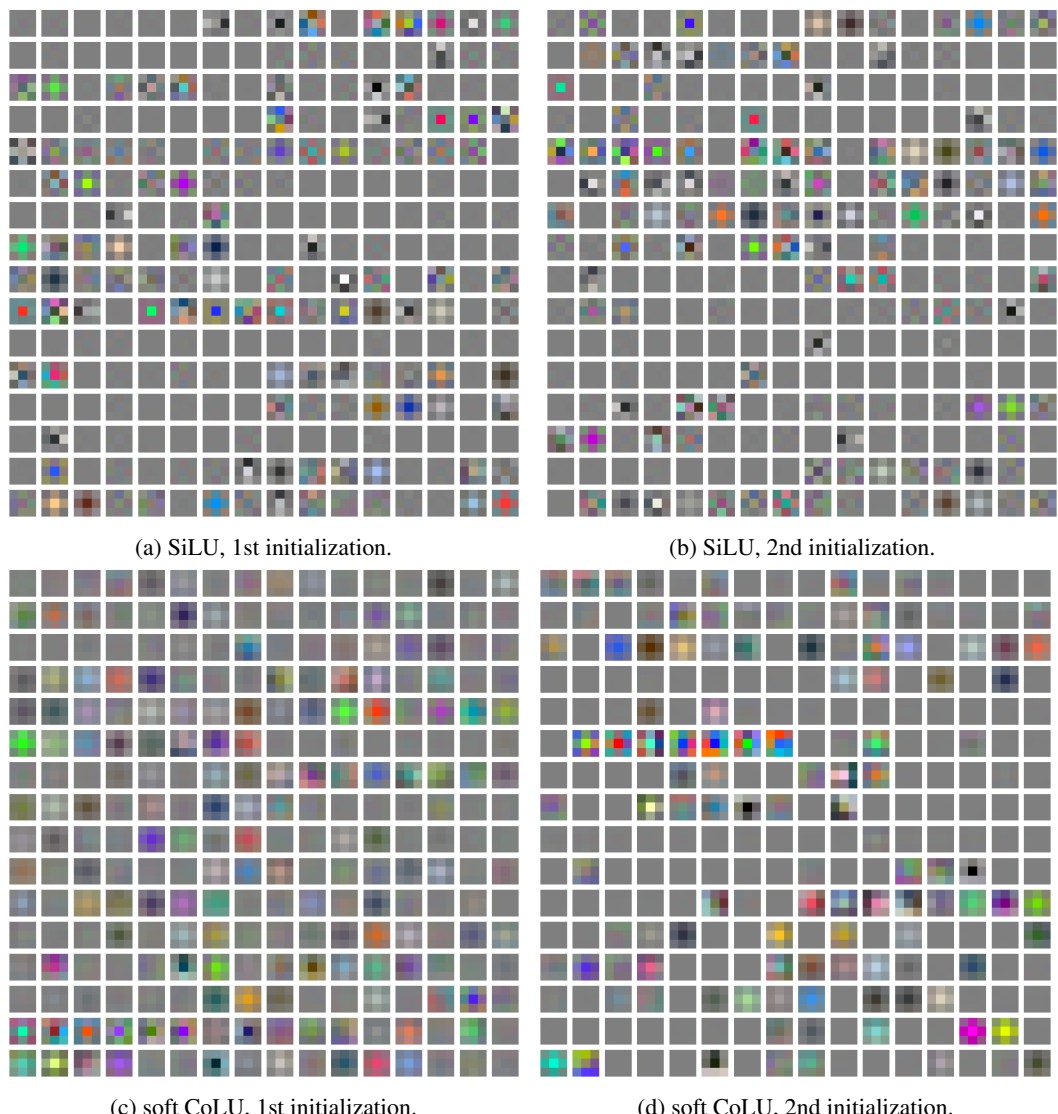

|  |  |
|---|---|
| (a) SiLU, 1st initialization. | (b) SiLU, 2nd initialization. |
| (c) soft CoLU, 1st initialization. | (d) soft CoLU, 2nd initialization. |

Figure 15: Palettes of diffusion models with SiLU and soft CoLU. The first row can be permuted to match each other whereas the second row cannot.

generative model) in a ReLU-model can be permuted to match each other, whereas a CoLU-model cannot, showing that the symmetric pattern is essentially different from permutation. The diffusion model implementation is based on [Salimans and Ho, 2022] and we only change the activation function to be conic with $G = 32$ without axis sharing. We take a global batch size of 128 and a learning rate of $10^{-4}$. After around 5K steps the generated images are perceptually visible. Figure 15 visualizes the last convolution layer $w$ (which we call a palette) of dimension $256 \times 3 \times 3 \times 3$ in SiLU model and soft CoLU model, each with two different initializations. The colors are linearly scaled for better visualization. The left two sets of filters can be permuted to match each other, whereas the right two sets cannot since they are orthogonal symmetric except for the axes. We observe that the last layer has more visually plausible patterns than the first layer in the denoising UNet, different from most works in the literature do for recognition models.

**Generative Model Alignment**   We show linear mode connectivity results for the same toy model in Section 5.2, and we find out that linear mode connectivity also holds in generative models, which is rarely discussed in the literature.

Weight matching and state matching algorithms in Appendix F are applied to align the VAE model, and the results are shown in Figure 12. They have different advantages: weight matching produces a flatter barrier in our toy experiment, and state matching requires no data as the model input. Their convergence is analyzed in [Ainsworth et al., 2023, Jordan et al., 2023]. The difference in the conic case is that the symmetry group is relaxed, so the Stiefel manifold optimization problem replaces the sum of bilinear assignment problem (SOBLAP). Figure 13 and 14 visualize the loss barrier and the KL Divergence barrier.

# E   Proofs

*Proof of Proposition 4.2.* If $|x_\perp| \neq 0$, Equation (6) holds component-wise, and the set $\{x \in \mathbb{R}^C : |x_\perp| = 0\}$ is negligible. $\qquad\square$

*Proof of Lemma 4.5.* We assume $P \in \mathrm{GA}(C)$ To prove $\mathcal{G}_\lambda \subset \mathcal{G}_{\mathcal{I}_\lambda}$, it suffices to show $\forall P \in \mathcal{G}_\lambda, \mathcal{I}_\lambda = PP^{-1}\mathcal{I}_\lambda = P\mathcal{I}_\lambda \subset I_\lambda$. The last inclusion comes from $\forall P \in \mathcal{G}$, there holds $\forall x \in \mathcal{I}_\lambda, \lambda(Px) = P\lambda(x) = Px$, so $Px \in \mathcal{I}_\lambda$. The first equality is from $P \in \mathcal{G}_\lambda$ and the second one is from $x \in \mathcal{I}_\lambda$. Conversely, to prove $\mathcal{G}_{\mathcal{I}_\lambda} \subset \mathcal{G}_\lambda$, we need to strengthen the condition on $\lambda$ to $\exists A$ a convex set such that $\forall x, \lambda(x) = \mathcal{P}_A(x)$. $\forall z \in \mathcal{I}_\lambda, \langle z - P\lambda(x), Px - P\lambda(x) \rangle \geq 0$, so $\lambda(Px) = P\lambda x$ $\qquad\square$

*Proof of Proposition 4.8.* (1) is proven by taking $x$ and $P$ such that

$$w'x = (1, 0, 0, \ldots, 0), \quad P[1, 2; 1, 2] = \begin{bmatrix} \frac{\sqrt{2}}{2} & -\frac{\sqrt{2}}{2} \\ \frac{\sqrt{2}}{2} & \frac{\sqrt{2}}{2} \end{bmatrix}$$

(2) is proven by taking

$$w^\dagger = \begin{bmatrix} 0 & \mathbf{I}_2 \\ \mathbf{I}_{C-2} & 0 \end{bmatrix}, \quad w'^\dagger = \begin{bmatrix} 0 & \mathbf{I}_{C-2} \\ \mathbf{I}_2 & 0 \end{bmatrix}$$

$\qquad\square$

*Proof of Remark 4.9.* It suffices to take $\eta$ large enough so that $D \in \mathrm{Diag}(C)$ is determined by $\mathrm{argmin}_{PD \in \mathcal{G}_\lambda} \|PD\theta\|$, since $P \in \mathrm{Perm}(C)$ does not change $\|P\theta\|$. $\qquad\square$

# F   Algorithms

Algorithm 1 and 2 from Jordan et al. [2023], Ainsworth et al. [2023] are applied to achieve linear mode connectivity of the toy VAE model.

# G   Unification of Neural Networks

This section aims to establish a bottom-up framework from first principles to infer the form of neural network architectures, including the proposed activation function. For simplicity, we assume that each state is defined in a vector space with a fixed dimension $M = \mathbb{R}^C$. We separate the construction into several parts, including a general Neural Network, a Residual Network, a Convolutional Network, and an Attention Network.

**Proposition G.1** (Derivation of a Neural Network). *The assumptions on the left of the following equations characterize the neural network in Equation (20).*

---

**Algorithm 1** Weight Matching

**Require:** $\theta_0, \theta_1$ ▷ Pre-Trained Weights from different random initializations
**Require:** $x \in X$ ▷ Intermediate states ordered by forward pass
**Require:** $\theta_{\text{prev}}(x), \theta_{\text{next}}(x)$ ▷ Linear weights prior to and after the state
**Require:** $\mathcal{G}$ ▷ Symmetry group of the activation function
**Ensure:** $P = \{P_x : x \in X\}$ ▷ Optimal alignment
 1: Initialize $P_x = \mathbf{I}_{\dim(x)}$ ▷ Identity matrices with the same dimension of $x$
 2: **repeat**
 3:     **for** $x$ in $\text{RandPerm}(X)$ **do** ▷ Shuffle the order of the states
 4:         $L(P) = 0$
 5:         **for** $w'$ in $\theta_{\text{prev}}(x)$ **do**
 6:             **for** $w$ in $\theta_{\text{next}}(x)$ **do**
 7:                 $L(P) \leftarrow L(P) + \text{tr}(w_0'^\top P w_1')/|\theta_{\text{prev}}(x)| + \text{tr}(w_0 P^\top w_1^\top)/|\theta_{\text{next}}(x)|$
 8:             **end for**
 9:         **end for**
10:         Solve $P_x \leftarrow \text{argmin}_{P \in \mathcal{G}} L(P)$
11:     **end for**
12: **until** $P$ Converges

---

**Algorithm 2** State Matching

**Require:** $\theta_0, \theta_1, \theta_{\text{prev}}(x), \theta_{\text{next}}(x), \mathcal{G}$ ▷ Same as above
**Require:** $x(0)$ ▷ Data as model input
**Require:** $x \in X(\theta, x(0))$ ▷ Following the order of the forward pass
**Ensure:** $P = \{P_x : x \in X(\theta, x(0))\}$ ▷ Optimal alignment
 1: Initialize $P_x = \mathbf{I}_{\dim(x)}$
 2: **for** $(x_0, x_1)$ in $(X(\theta_0, x(0)), X(\theta_1, x(0)))$ **do**
 3:     Solve $P_\ell \leftarrow \text{argmin}_{P \in \mathcal{G}} L(P) = x_0^\top P x_1$
 4:     **for** $w'$ in $\theta_{\text{prev}}(x)$ **do**
 5:         $w_1 \leftarrow P_\ell w_1$
 6:     **end for**
 7:     **for** $w$ in $\theta_{\text{next}}(x)$ **do**
 8:         $w_1 \leftarrow P_\ell w_1$
 9:     **end for**
10: **end for**

---

$$x(1) = \Lambda(x(0)) \tag{15}$$

$$\stackrel{\text{Process Decomposition}}{\Longrightarrow} x(L) = \Lambda_L \Lambda_{L-1} \ldots \Lambda_1(x(0)) \tag{16}$$

$$\stackrel{\text{Linear Kernel Space}}{\Longrightarrow} x(L) = w(L)\Lambda_L(w'(L) \ldots w(1)\Lambda_1(w'(1)(x(0)) \ldots)) \tag{17}$$

$$\stackrel{\text{Time Homogeneity}}{\Longrightarrow} x(L) = w(L)\Lambda(w'(L) \ldots w(1)\Lambda(w'(1)(x(0)) \ldots)) \tag{18}$$

$$\stackrel{\text{Component-Wise}}{\Longrightarrow} x(L) = w(L)\lambda(w'(L) \ldots w(1)\lambda(w'(1)(x(0)) \ldots)) \tag{19}$$

$$\stackrel{\text{Iterative Form}}{\Longleftrightarrow} x(\ell) = w(\ell)\lambda(w'(\ell)x(\ell-1)), \ell = 1, 2, \ldots L \tag{20}$$

In the derivation above, equation (15) denotes an arbitrary function $\Lambda$ with input $x(0)$ and output $x(1)$. Equation (16) holds by assuming the function decomposes into several ones, resulting in a process or a sequence of states $x(0), x(1), \ldots, x(L) \in M$, where the terminal states $x(0)$ and $x(L)$ are the input and output. Equation (17) follows from assuming the sequence of functions to perform in a linear kernel space. Suppose the linear kernel function at layer $\ell$ parameterized by $w'$

$$\Phi : M \longrightarrow M_\lambda$$
$$x \longmapsto w'(\ell)x$$

on the kernel space, there is a function $\Lambda : M_\lambda \to W_\lambda$ where $W_\lambda$ is the range of the activation function. Then the inverse kernel function is parameterized by $w$

$$\widehat{\Phi} : W_\lambda \longrightarrow M$$
$$x \longmapsto w(\ell)x$$

Again we assume for simplicity that the dimensionality of each kernel space is fixed: $M_\lambda = M = \mathbb{R}^C$. Equation (17) is obtained by replacing $x \in M$ with $x' \in M_\lambda$ in equation (16) and plugging in the change of variable $x' = wxw'$. Equation (18) is obtained by assuming *time homogeneity modulo a linear group* of the nonlinear functions: the function $\Lambda$ is on the lifted space $M_\lambda$ in Equation (17) instead of $M$, where the lifting is determined by assuming that there exist proper $w, w'$ in each space such that the functions are uniform over time, meaning $\Lambda_1 = \Lambda_2 = \ldots = \Lambda_L = \Lambda$. Equation (19) assumes that there exists a function $\lambda : \mathbb{R} \to \mathbb{R}$ so that the nonlinear function is represented as $\Lambda(x_1 e_1 + \ldots + x_n e_n) = \lambda(x_1)e_1 + \ldots + \lambda(x_n)e_n$. In this paper, we replace this assumption with orthogonal symmetry instead. Note that the component-wise $\lambda : M_\lambda \to M_\lambda$ is equivariant under any permutation $P$. Equation (20) rewrites the process into steps between adjacent states.

**Proposition G.2** (Derivation of a Residual Network). *Adding more assumptions, we continue to derive the form of a Residual Network in Equation (27).*

$$\stackrel{\textit{Linear Splitting}}{\Longleftrightarrow} \quad x(\ell) = \lambda(w'(\ell)x(\ell-1)) + (w(\ell)-1)\lambda(w'(\ell)x(\ell-1)) \tag{21}$$

$$\stackrel{\textit{Re-Parameterization}}{\Longrightarrow} \quad x(\ell) = \lambda(w'(\ell)x(\ell-1)) + w(\ell)\lambda(w'(\ell)x(\ell-1)) \tag{22}$$

$$\stackrel{\textit{Linear Branching}}{\Longrightarrow} \quad x(\ell) = \lambda(w''(\ell)x(\ell-1)) + w(\ell)\lambda(w'(\ell)x(\ell-1)) \tag{23}$$

$$\stackrel{\textit{Nonlinear Branching}}{\Longrightarrow} \quad x(\ell) = \lambda'(w''(\ell)x(\ell-1)) + w(\ell)\lambda(w'(\ell)x(\ell-1)) \tag{24}$$

$$\stackrel{w''=1}{\Longrightarrow} \quad x(\ell) = \lambda'(x(\ell-1)) + w(\ell)\lambda(w'(\ell)x(\ell-1)) \tag{25}$$

$$\stackrel{w'=1}{\Longrightarrow} \quad x(\ell) = \lambda'(x(\ell-1)) + w(\ell)\lambda(x(\ell-1)) \tag{26}$$

$$\stackrel{\textit{Residualization}}{\Longrightarrow} \quad x(\ell) = x(\ell-1) + w(\ell)\lambda(x(\ell-1)) \tag{27}$$

In the derivation above, Equation (21) splits the inverse kernel function's weight $w$ into the identity (zeroth-order) part and the first-order part $w-1$. Equation (22) re-parameterize the weights by denoting $1-w$ as $w$ without loss of generality. Equation (23) modifies the assumption in Equation (17) so that two copies of kernel functions are parameterized by $w'', w'$, and the inverse kernel function remains the same. Equation (24) modifies the assumption in equation (18) different functions $\lambda', \lambda$ applies on each one. Equation (25) assumes that the first kernel function $w''$ is identity. Equation (26) further assumes $w'$ is identity to simplify equations in the sequel. Equation (27) assumes that the function associating to the zeroth-order kernel space is identity.

**Proposition G.3** (Derivation of a Convolutional Network). *Given a basic neural network, the form of a convolutional neural network in Equation (33) is determined by the following additional assumptions on the left.*

$$\stackrel{\textit{Space Indexation}}{\Longleftrightarrow} \quad x(\ell) = x(\ell-1) + w(\ell,\omega,\omega',\sigma,\sigma')\lambda(x(\ell-1,\omega',\sigma')) \tag{28}$$

$$\stackrel{\textit{Summation Form}}{\Longleftrightarrow} \quad x(\ell) = x(\ell-1) + \sum_{\omega' \in \Omega} w(\ell,\omega,\omega',\sigma,\sigma')\lambda(x(\ell-1,\omega',\sigma')) \tag{29}$$

$$\stackrel{\textit{Equivariance}}{\Longrightarrow} \quad x(\ell) = x(\ell-1) + \sum_{\omega' \in \mathbb{Z}^2} w(\ell,\omega'-\omega,\sigma,\sigma')\lambda(x(\ell-1,\omega',\sigma')) \tag{30}$$

$$\stackrel{\textit{Change of Variable}}{\Longleftrightarrow} \quad x(\ell) = x(\ell-1) + \sum_{\omega' \in \mathbb{Z}^2} w(\ell,\omega',\sigma,\sigma')\lambda(x(\ell-1,\omega'+\omega,\sigma')) \tag{31}$$

$$\stackrel{3\times 3\ \textit{Window}}{\Longrightarrow} \quad x(\ell) = x(\ell-1) + \sum_{\omega \in \{-1,0,1\}^2} w(\ell,\omega+\omega',\sigma,\sigma')\lambda(x(\ell-1,\omega',\sigma')) \tag{32}$$

$$\stackrel{\textit{Convolution Notation}}{\Longleftrightarrow} \quad x(\ell) = x(\ell-1) + w(\ell) \star \lambda(x(\ell-1)) \tag{33}$$

In the derivation above, Equation (28) stacks the states of dimension $n = CHW$ into a tensor whose space dimensions is indexed by $\omega \in \Omega = [H] \times [W] \subset \mathbb{Z}^2$ (with the bracket notation $[n] = \{1, 2, \ldots, n\}$) and the channel dimension indexed by $\sigma \in [C]$. Equation (29) write the matrix-vector product in the form of a summation. In Equation (30) we imposes core assumption of the Convolutional Neural Network, namely the spatial translation equivariance, so that $(wx)(\omega - \omega'') = (wx(\omega - \omega''))$, $\forall \omega''$. This results in $w(\omega, \omega' + \omega'') = w(\omega - \omega'', \omega')$, $\forall \omega''$, so $w(\omega, \omega')$ must take the form of $w(\pm\omega \mp \omega')$, and we set $w(\omega' - \omega)$ without loss of generality. Equation (31) is a change of variable, replacing $\omega' - \omega$ with $\omega'$. Equation (32) imposes the condition that the spatial dependency on $\Omega$ is within a $3 \times 3$ neighbourhood. Note that the family of neighbourhoods defines the *Topology* of the space $\Omega$. Finally, Equation (33) denotes the linear function with the $\star$ notation.

**Proposition G.4** (Derivation of an Attention Network)**.** *The construction of the cross-attention function is proceeded by imposing further assumptions.*

$$\stackrel{1 \times 1 \text{ Window}}{\Longrightarrow} \quad x(\ell) = x(\ell - 1) + \lambda(x(\ell - 1, \omega, \sigma'))w(\ell, \sigma', \sigma) \tag{34}$$

$$\stackrel{\text{Condition } k^T k}{\Longrightarrow} \quad x(\ell) = x(\ell - 1) + \lambda(x(\ell - 1, \omega, \sigma'))k(\ell, \sigma'', \sigma')^T k(\ell, \sigma'', \sigma')w(\ell, \sigma', \sigma) \tag{35}$$

$$\stackrel{\lambda=1}{\Longrightarrow} \quad x(\ell) = x(\ell - 1) + x(\ell - 1, \omega, \sigma')k(\ell, \sigma'', \sigma')^T k(\ell, \sigma'', \sigma')w(\ell, \sigma', \sigma) \tag{36}$$

$$\stackrel{\text{Scaling}}{\Longrightarrow} \quad x(\ell) = x(\ell - 1) + \text{softmax}(x(\ell - 1, \omega, \sigma')k(\ell, \sigma'', \sigma')^T)k(\ell, \sigma'', \sigma')w(\ell, \sigma', \sigma) \tag{37}$$

$$\stackrel{Q,K,V \text{ Notations}}{\Longleftrightarrow} \quad x(\ell) = x(\ell - 1) + w(\ell)\,\text{softmax}(QK^T)V \tag{38}$$

In the above derivation, Equation (34) assumes the Topology to be discrete, or the neighbourhood of a spatial point is itself, which restricts the convolution to be on a $1 \times 1$ window. For the matrix $w(\ell, \sigma, \sigma')$ with $\sigma, \sigma' \in [C]$, Equation (35) applies the linear transform $k^T k$, where $k(\ell, \sigma'', \sigma)$ can be regarded as a set of $C''$ condition "pixels" of dimension $C$, or $\omega \in [C], \omega'' = [C''']$. Equation (36) assumes $\lambda$ to be identity function denoted as 1. Equation (37) scales $xw^T$ with a softmax function $\text{softmax}(x(\sigma, \sigma'')) = \exp(x(\sigma, \sigma''))/\sum_{\sigma'' \in [C'']} \exp(x(\sigma, \sigma''))$. Finally, Equation (37) is obtained from setting the Query-Key-Value notations $Q = x(\ell - 1, \omega, \sigma'), K = V = k(\ell, \sigma', \sigma'')$. Note that by cancelling the assumption in Equation (26), we may also take in $Q = w_Q(\ell, \sigma, \sigma')Q', K = w_K(\ell, \sigma, \sigma')K', V = w_V(\ell, \sigma, \sigma')V'$.

**Proposition G.5** (Attention Network Dynamics)**.**

$$\stackrel{\text{Attention Dynamics}}{\Longrightarrow} \quad \dot{x} = \text{softmax}(QK^T)V \tag{39}$$

Equation (39) is obtained by setting $w(\ell)$ as identity and consider $\ell \in [0, L]$.

**Proposition G.6** (ResNet Dynamics)**.** *By assuming the continuation $\ell \in [0, L]$, we obtain the continuous dynamics of ResNet*

$$\stackrel{\text{ResNet Dynamics}}{(39) \Longrightarrow} \quad \dot{x} = \lambda(x) \tag{40}$$

