# OpenReview forum: "Conic Activation Functions"
_NeurIPS.cc/2024/Workshop/UniReps — UniReps_

### Official Review · Reviewer_1jGz · 2024-10-05

**Rating:** 7
**Confidence:** 4

**Review:**

The paper presents a novel activation function called Conic Activation Functions (CoLU), which generalizes neural networks' symmetry from permutations to continuous orthogonal groups. This new activation is inspired by geometric principles, specifically the Lorentz cone, and is proposed as an alternative to traditional component-wise activation functions like ReLU. The core idea is that CoLU introduces new forms of symmetry into neural networks, which can improve performance in various tasks.
The authors demonstrate that CoLU outperforms ReLU across multiple architectures such as MLPs, ResNet, Transformers, and UNet, particularly in image/text classification and generation. They further explore enhancements like multi-head structure, soft scaling, and axis sharing to improve performance.

Strengths:
1. The method is novel. The introduction of a non-component-wise activation function (CoLU) that generalizes neural networks symmetry to continuous orthogonal groups is a fresh approach.

2. Good performance on various architectures is offered by CoLU shows significant improvements over ReLU in various deep learning models (MLP, ResNet, Diffusion Models).

3. The idea seems mathematically sound to me, and makes sense.

Weaknesses:
1. The paper can be better organised and written. Its overall flow can be improved. It is also not clear what is the purpose of the first paragraph in the Introduction section.

2. The authors should discuss the computational cost aspect of this method.

3. While the authors do consider many experiments, the method can be better benchmarked and compared with more architectures and recent activation functions, especially learned activation functions. (see question below)

Question to authors:
I think that the idea of multi-component activation function makes sense. Can you discuss or compare with the recent approach in [1] that offers a component-wise activation that is dependent on all components? would such an approach benefit Conic Activation Functions?

[1] DiGRAF: Diffeomorphic Graph-Adaptive Activation Function

---

### Official Review · Reviewer_Y36Q · 2024-10-05
**Conic activation functions**

**Rating:** 7
**Confidence:** 4

**Review:**

The paper introduces Conic Activation Functions (CoLU), a new type of activation function for neural networks that generalizes the symmetry principles of component-wise activations like ReLU. The main motivation is to extend the symmetry in neural networks beyond permutations by leveraging continuous orthogonal groups, thus providing an alternative symmetry structure based on the Lorentz cone. CoLU maintains a similar computational cost as traditional activations but shows superior performance across some toy model architectures. Experimental results seem to show improved loss and training speed, particularly when low-dimensional cones are used, along with configurations like multi-head structures and axis sharing.

Strengths:
- The proposed methodology is quite interesting and new.
- The proposed approach can be effectively used on different architectures.
- The toy tests seem to show an improvement for models performances.
-  It's very interesting how the paper proposes geometric and algebraic symmetry for network design.

Weaknesses:
- While the paper claims that the computational complexity remains the same as ReLU, it doesn't provide an analysis of the overhead in large-scale networks, especially when using more complex configurations like multi-head structures.
- The paper seems quite strong on the theoretical side, but it might lack some practical consideration aside from the model performances.

---

### Official Review · Reviewer_qTG8 · 2024-10-06

**Rating:** 6
**Confidence:** 3

**Review:**

**Summary:**

The paper introduces Conic Activation Functions (CoLU) to enhance symmetry in neural networks beyond standard permutations. Inspired by Lorentz cones, CoLU demonstrates superior performance compared to ReLU and SiLU across various architectures like ResNets, UNets, and Transformers, in both image and text tasks. The authors explore variations such as soft scaling, multi-head structures, and axis sharing, which further enhance CoLU’s efficacy in reducing loss and speeding up training.

**Strengths and Weaknesses:**


**Strengths:**

- The paper proves that CoLU induces orthogonal symmetries in the latent space, potentially improving generalization. This aligns with findings from (Zhao et al., 2023b), where such symmetries are linked to flatter minima, which is often associated with better model robustness.

- CoLU can be applied selectively to subsets of neurons rather than globally, making it particularly useful in architectures like Regular Group Convolutions, where preserving equivariance by restricting cross-channel mixing is crucial.

- CoLU's superior performance compared to ReLU and SiLU across diverse tasks is well-demonstrated. Additionally, the connection between CoLU and physics-based concepts enhances the potential for interpretability, which could be an exciting avenue for future research.

**Weaknesses:**

- While the concept of CoLU is relatively straightforward, its current presentation is overly complex, which could hinder accessibility for a broader audience. Simplifying the notation and adhering to established conventions from weight symmetry literature would enhance readability. Additionally, providing geometric visualizations (e.g., a 3D illustration of the conic projection) would make the concept more intuitive.

- Section B of the Appendix, though interesting, appears tangential to the core contribution of the paper and might dilute the focus. It may be worth revising or relocating this content to ensure relevance.

- Other activation functions, such as the radial scaling activation (Ganev et al., 2022), also provide orthogonal symmetries in latent spaces. Since the symmetry groups are similar when $G=1$, these functions should be acknowledged and compared in the experimental section for completeness.

- Corollary 4.6 addresses the monomial matrix group as the intertwiner group (notation from Godfrey, Charles, et al., 2022) for certain point-wise activation functions. However, if the activation is homogeneous, the diagonal matrix should be constrained to positive values. Similarly, for `tanh`, the matrix entries are restricted to $\pm 1$ (Tran, Hoang V., et al., 2024). A more precise statement in Corollary 4.6 is necessary to account for these nuances.

**Questions:**

- Could you elaborate on your claim that “CoLU considers finer symmetries than GCNN”? Specifically, what finer symmetries does CoLU capture, and how do they compare to those modeled by GCNN?

- Could you provide more detail regarding the experiment in Section 5.6, where you fine-tune a large diffusion model with component-wise activation functions towards its conic version? What insights or implications do you draw from this approach?

**Limitations:** Limitations are adequately addressed

**Ethical concerns:** No

**Soundness:** 2 fair

**Presentation:** 1 poor

**Contribution:** 3 good

---

### Official Review · Reviewer_Wxn2 · 2024-10-07
**Big on theoretical motivation, short on emperical results**

**Rating:** 4
**Confidence:** 3

**Review:**

Strengths:
- The main theoretical motivation of the proposed activation function is strong: specifically that permutation symmetry and the work of Ainsworth et al. specifically, suggests NN optimization is approx. convex modulo a group. This motivation got me excited about the research direction.
- Well-written paper overall.
- Good background, covers everything needed to understand the work well.
- Excellently written and presented method section, specifically Figure 1 which I believe explains the method well.

Weaknesses:
- At the end of the day, no matter how beautiful the math, proposing a new activation function has a significant bar to pass empirically: it has to be demonstrated to work significantly better empirically than existing activation functions at least in some general setting across a range of datasets/models. To the author's credit, they have evaluated their proposed method over a variety of settings including image classification, diffusion models, and the MLP layer of NLP transformers. However, overall I did not find convincing evidence of a substantial improvement in any of the results. furthermore, the only compared "component-wise" activation function is ReLU, and there are obviously a lot more of them out there (as discussed in background)
I would encourage the authors to focus on a setting where their proposed method shows the biggest advantage and then show as many results across variants of models/datasets in that setting in comparison to a variety of existing activation functions.
   - Toy MLP/MNIST result. 95% on MNIST is an extremely poor baseline already even for an MLP, but to be fair the authors do call this a toy setting.
   - ResNet-56/CIFAR10: ResNet-56 (ReLU) test accuracy on CIFAR-10 in the original paper is 93.03%, so again I'm not sure why this baseline is so low. If it wasn't for that, I think this would be the strongest result in the paper (although not sufficient alone), and I would encourage the authors to improve upon the baseline and perhaps show results on ResNet with other datasets and component-wise activations for more convincing results.
  - UNet training loss differs by 0.0013, I don't believe this is significant although we don't have variance to judge with.
  - GPT2 MLP result shows a slightly higher training loss and virtually equivalent evaluation loss (differs by 0.001), I don't believe this is significant although we don't have variance to judge with.
- Figure 5 does not explain which "component-wise" activation function is being plotted.
- Give the empirical results, it makes myself at least wonder after the fact, indeed if NN optimization is approx. convex module a group, then doesn't training from any specific random init mean it's already approximately convex. Perhaps having the explicitly connected-local minima of the proposed conic activation function doesn't help if you're virtually guaranteed to be within a good one already? Perhaps outside of convincing empirical results, some linear-mode connectivity or other loss-based analysis would be more convincing that the resulting loss is in fact easier to optimize on?
- Notation $x_{-1}$ looks more like a typo at a glance than normal mathematical notation, would suggest changing this.
- Minor: Tables/figures should in general be at top/bottom of page rather than interspersed in text.

---

### Decision · Program_Chairs · 2024-10-10

**Decision:**

Accept

**Comment:**

In light of the positive reviewers' feedback and relevancy of the submission, we are pleased to accept this paper for presentation at UniReps 2024. We kindly ask the authors to incorporate the reviewers' suggestions and feedback in the final camera-ready version of the manuscript.